# Comparative Analysis of Transposable Elements in Strawberry Genomes of Different Ploidy Levels

**DOI:** 10.3390/ijms242316935

**Published:** 2023-11-29

**Authors:** Keliang Lyu, Jiajing Xiao, Shiheng Lyu, Renyi Liu

**Affiliations:** 1College of Horticulture, Fujian Agriculture and Forestry University, Fuzhou 350002, China; lvkelianglkl@163.com (K.L.); kingguoguo@163.com (S.L.); 2Haixia Institute of Science and Technology, Fujian Agriculture and Forestry University, Fuzhou 350002, China; jjxiao@fafu.edu.cn

**Keywords:** *Fragaria*, cultivated strawberry, transposable elements (TEs), genome evolution, long terminal repeat retrotransposons (LTR-RTs)

## Abstract

Transposable elements (TEs) make up a large portion of plant genomes and play a vital role in genome structure, function, and evolution. Cultivated strawberry (*Fragaria x ananassa*) is one of the most important fruit crops, and its octoploid genome was formed through several rounds of genome duplications from diploid ancestors. Here, we built a pan-genome TE library for the *Fragaria* genus using ten published strawberry genomes at different ploidy levels, including seven diploids, one tetraploid, and two octoploids, and performed comparative analysis of TE content in these genomes. The TEs comprise 51.83% (*F. viridis*) to 60.07% (*F. nilgerrensis*) of the genomes. Long terminal repeat retrotransposons (LTR-RTs) are the predominant TE type in the *Fragaria* genomes (20.16% to 34.94%), particularly in *F. iinumae* (34.94%). Estimating TE content and LTR-RT insertion times revealed that species-specific TEs have shaped each strawberry genome. Additionally, the copy number of different LTR-RT families inserted in the last one million years reflects the genetic distance between *Fragaria* species. Comparing cultivated strawberry subgenomes to extant diploid ancestors showed that *F. vesca* and *F. iinumae* are likely the diploid ancestors of the cultivated strawberry, but not *F. viridis*. These findings provide new insights into the TE variations in the strawberry genomes and their roles in strawberry genome evolution.

## 1. Introduction

Transposable elements (TEs) are mobile and repetitive DNA sequences dispersed throughout the genomes of most eukaryotes [1]. First identified in *Zea mays* (maize) [2], TEs make up the majority of genetic material in plant genomes [3]. For instance, TEs comprise approximately 85% of the maize genome [3]. Two classes of mobile genetic elements, RNA TEs and DNA TEs, possess the capability to replicate and relocate within the nuclear genome when they are active and even transfer horizontally across diverse species [4]. In addition, TEs can perturb host sequences and recombine homologously, causing DNA rearrangements [5], which significantly impact the genome structure and function. Recent investigations have shown that a long terminal repeat retrotransposon (LTR-RT) was inserted in the upstream of *MdMYB1*, resulting in the conversion of golden-skinned apples to red-skinned apples [6]. Throughout maize domestication, numerous TEs have increased alternative splicing levels in transcription factors and stress-responsive genes [7]. Furthermore, TEs exhibit remarkable rates of amplification and removal, thereby driving changes in genome size and the modification of agronomic traits [6,8,9,10]. Therefore, their significance in the realm of plant genome structure, evolution, and function is substantial.

As a crucial and abundant part of plant genomes, TEs have been used as genetic markers in plant research and breeding. Molecular marker systems based on TEs have been effectively employed in crop breeding, notably in rice and tea, enabling precise identification of desirable traits and the development of superior crops [11,12]. The use of TE-derived markers in genome-wide association studies (GWAS) may recover a large portion of single-nucleotide polymorphism (SNP)-based GWAS peaks and, in the meantime, reduce false positives associated with linkage disequilibrium among SNP markers [13]. Therefore, it is critical to carefully annotate TEs in plant genomes and perform comparative analysis of TE diversity across different species [14,15,16,17].

Strawberry belongs to the genus *Fragaria* of the Rosaceae family and is one of the most important fruit crops in the world. The cultivated strawberry (*F. x ananassa*) is an allo-octoploid species developed in Europe in the mid-18th century through hybridization between two wild octoploid ancestors (*F. chiloensis* and *F. virginiana*) [18,19,20] and contains four different subgenomes. It is widely accepted that *F. vesca* is a diploid ancestor that bears the closest genetic distance to the cultivated strawberry, and it became part of the octoploid strawberry genome around 1.1 million years ago [18,19,20,21]. Nonetheless, there remains substantial debate regarding the other three diploid ancestors. Edger and colleagues suggest that the other three subgenomes were provided by three different diploid ancestors, namely, *F. iinumae*, *F. nipponica*, and *F. viridis* [20]. However, it was also proposed that all three subgenomes were contributed solely by *F. iinumae* or its ancestor [22,23]. Thus, a consensus on the diploid ancestors of cultivated strawberry has not been reached, and comparative analysis of TEs in different strawberry genomes may provide a clue.

TEs are a major genetic component in strawberry genomes and may play a role in the rich phenotypic diversity of strawberry species. The *Fragaria* genus comprises 25 identified species that are widely distributed throughout the Northern Hemisphere [18,24,25]. *Fragaria* species display rich diversity in phenotypes such as disease resistance, cold tolerance, fruit color, and flavor [26,27,28], making them valuable germplasm resources for the breeding of cultivated strawberry [29,30]. In previous studies on strawberry genomes, researchers identified over 40% of TE content in each genome [20,21,25,28], but the annotation and analysis of TEs remain limited. TEs are also critical players in shaping phenotypic traits of strawberries. For example, an LTR-RT inserted into the genome of *F. nilgerrensis* leads to the white fruit phenotype [28,31]. In cultivated strawberry, TE-derived long non-coding RNAs (lncRNAs) are involved in strawberry growth, development, and fruit maturation through the ABA biosynthesis pathway [32]. Therefore, exploring the roles of TEs in strawberry diversity and their impact on phenotypes is an important research area.

The publication of numerous high-quality genomes of *Fragaria* species [20,21,22,25,28,33,34,35] provides a golden opportunity for comparative analysis of TEs in the *Fragaria* genomes. However, a systematic analysis of TEs in species of the *Fragaria* genus is still lacking. Here, we utilized ten published high-quality genomes in the *Fragaria* genus to construct a pan-genome TE library and performed a comprehensive identification and analysis of the TEs. The results show that TEs are an important force in shaping the strawberry genomes, and the copy numbers of different LTR-RT families reflect the genetic distance among species. Furthermore, comparison of TEs within the subgenomes of cultivated strawberry with those in their diploid ancestors suggests that *F. vesca* and *F. iinumae*, but not *F. viridis*, likely served as the diploid ancestors of cultivated strawberry. Therefore, this study provides new insights on the TE landscape in the strawberry genomes at different ploidy levels as well as the evolutionary history of the strawberry genomes. The TE data that we generated can also serve as an invaluable resource for investigating strawberry genome structure and evolution and for guiding strawberry breeding and phenotypic trait improvements.

## 2. Results

### 2.1. Construction of a Pan-Genome TE Library with Ten Fragaria Species

#### 2.1.1. Pan-Genome TE Content across Genus *Fragaria*

Transposable elements (TEs) were identified in the genomes of ten *Fragaria* species at different ploidy levels, including seven diploids, one tetraploid, and two octoploids, utilizing a de novo approach. A total of 43,648 consensus TE sequences were identified using EDTA [36], with each genome yielding between 3091 and 8001 consensus sequences (Appendix A). The *F. chiloensis* and *F. x ananassa* octoploid strawberries possess four subgenomes, leading to a higher count of TEs compared to other strawberries that have a lower ploidy level. After discarding simple repeat sequences and those less than 80 nt in length, the pan-genome TE library of strawberries was established by removing highly similar sequences from all TE sequences that were identified from each of the ten strawberry species (Figure 1). The library comprises a total of 26,226 consensus sequences (Table 1). Within this library, 7719 retrotransposons (class I) and 14,857 DNA transposons (class II) were documented (Table 1). Notably the *Fragaria* species exhibited a higher level of diversity among DNA transposons.

The TE annotation in the *Fragaria* species shows that TEs account for over 50% of the genome in every species (Figure 2 and Appendix A). Noteworthy variations in TE content were observed, with percentages ranging from 51.83% in *F. viridis* to 60.07% in *F. nilgerrensis*. Three diploid strawberries (*F. mandschurica*, *F. viridis*, *F. vesca*), residing in high-latitude regions and the southern Himalayas [25], exhibited lower TE content, with a rate of 52.77%, 51.83%, and 52.56%, respectively, compared to other strawberry species (55.34–60.07%), suggesting that higher latitude or lower temperature may reduce the transposition activity of TEs. On the other hand, polyploid strawberry species like *F. pentaphylla* and *F. chiloensis* had a similar TE content as that of the diploid strawberries that do not reside in high-latitude regions, ranging from 55.34% to 57.55%, suggesting polyploidization did not modify the TE content. As expected, pseudochromosome regions with a higher TE content tended to have a lower gene density (Appendix A).

#### 2.1.2. Diversity of TE Content in the Genome of Strawberries

Across all genomes, *F. iinumae* exhibited the highest proportion of LTR content (34.19%) among all TE classes (or subclasses), whereas TIR (14.33%) and Helitron (7.52%) contents were slightly lower than those in the other strawberry genomes (Figure 2 and Appendix A). In contrast to *F. iinumae*, the remaining strawberry genomes displayed a lower class I proportion, varying from 21.10% (*F. mandschurica*) to 27.46% (*F. chiloensis*), and a higher class II proportion, ranging from 26.37% (*F. x ananassa*) to 32.29% (*F. nilgerrensis*). In *F. iinumae*, the contents of various TEs superfamilies were marginally lower than those in other strawberry species, except that it had a substantially higher proportion of *gypsy* LTR retrotransposons (25.11%), suggesting an extensive amplification of this type of TE. The distinct TE profile of *F. iinumae* suggests that it has a unique TE amplification history because it was the earliest divergent diploid strawberry with a confined distribution in Japan and thus was geographically segregated from other strawberry species [25].

### 2.2. The Evolution History of TEs in the Fragaria Genus

#### 2.2.1. Amplification of TE Subfamilies

Because most TEs become inactive once inserted and are subsequently subject to random mutations, the insertion time (or age) of each TE can be estimated with the Kimura 2-parameter sequence divergence between each TE and the corresponding consensus sequence. Recently inserted TEs have a lower Kimura distance, whereas old TEs have a higher Kimura distance. Therefore, graphs can be generated for different kinds of transposons, i.e., LTR and LINE retrotransposons and TIR and Helitron DNA transposons in each strawberry genome, which depict peaks of activity that mark the occurrence of transposition bursts during the evolution of strawberries (Figure 3).

The results indicate that the TE transposition activities in all strawberry genomes were generally comparable, with only minor variations. A clear recent transposition burst event was observed, with the peak located in the range of a Kimura distance level of 4–7 (Figure 3). Notably, DNA transposons predominated in strawberries, except for *F. iinumae*. The genome of *F. iinumae* was found to be predominantly composed of retrotransposons. A transposition burst of LTR retrotransposons was also observed in the genomes of two diploid strawberries, *F. nilgerrensis* and *F. iinumae*, which did not correspond to any increase in copy number of DNA transposons. Four transposition bursts with a Kimura distance greater than 10 were detected in the genome of *F. vesca* with a low level of TE consent. Transposition activities in the two octoploid strawberry genomes were almost identical because the two species diverged no more than 500 years ago [20,21]. A more recent transposition burst event was detected in their genomes, in addition to the major transposition burst shared by all strawberries.

#### 2.2.2. TE Contents Are Strongly Correlated with Genome Sizes of Diploid Strawberries

Excluding the limited (only three) polyploid (tetraploid and octoploid) strawberry genomes, we tested the relationship between TE content and genome size in the remaining seven diploid strawberry genomes to examine the impact of TE content on strawberry genome size. We found a strong positive correlation between TE content and strawberry genome size (Figure 4A and Appendix A). This correlation was statistically verified by a Pearson correlation test (Pearson correlation coefficient: 0.75, *p* = 0.0528). The quantitative traits among species within the same phylogenetic branch are interdependent because of a shared ancestor. Therefore, we conducted phylogenetically independent contrasts (PICs) using phytools, based on a published phylogenetic tree [25,37]. After conducting PICs, we found a stronger positive correlation between TE content and diploid strawberry genome size, with a Pearson correlation coefficient of 0.85 (*p* = 0.0323). Therefore, transposable elements play a considerable role in determining the size of diploid strawberry genome.

#### 2.2.3. Removal Rate of LTR-RTs and the Strawberry Genome Size

LTR retrotransposons possess two identical terminal repeats on both ends, and sequence divergence between the two LTRs can be used to estimate the insertion time for each LTR-RT. We thus calculated the insertion time for all full-length LTR-RTs in each strawberry genome and put them in insertion time bins of 500 k years (Appendix A). Naturally, there was a declining trend in the graphs because old copies were constantly disrupted or removed from the genome (Appendix A).

Assuming that TE sequences are removed from the genome at a constant rate, the insertion time of TE distribution can be described by an exponential function with a constant half-life rate [38]. Our calculation results suggest that LTR retrotransposons are removed or truncated from the strawberry genomes at a rate of over 2.36 Mys (million years), with a notable variation between 2.36 Mys (*F. pentaphylla*) and 3.74 Mys (*F. nilgerrensis*) (Appendix A).

We analyzed seven diploid strawberry species (Figure 4B) to investigate the relationship between LTR-RT half-life rate and genome size and found a significant positive correlation between half-life rates and genome size, supported by a Pearson correlation test (Pearson correlation coefficient: 0.85, *p* = 0.0334). With PIC adjustment, a stronger positive correlation was also found (Pearson correlation coefficient: 0.93, *p* = 7.6 × 10^−3^). These results suggest that a slower removal rate could lead to an increase in genome size and that LTR-RT removal plays an important role in determining the genome size of diploid strawberries.

#### 2.2.4. The Transposition Bursts of LTR-RTs in Fragaria Genomes

The specific structure of LTR-RTs enables precise estimation of their transposition time through the genetic distance between the long terminal repeat sequences located at both ends. We computed the insertion time of intact LTR-RTs in each *Fragaria* genome, including 413–2103 *copia* elements and 155–2425 *gypsy* elements. A density plot of LTR-RT insertion times may reveal the transposition history of each LTR-RT family and transposition burst, especially for two main LTR-RT superfamilies, *copia* and *gypsy* (Figure 5A,B).

It is apparent that all strawberry genomes experienced a significant transposition burst in the recent past (Figure 5). The *F. nilgerrensis* genome appeared to have experienced two transposition bursts of *copia* elements, one about one million years ago and the other more than eight million years ago (Figure 5A). In comparison to other varieties of strawberries, recent *copia* transpositions (about 400,000 years ago) in the genomes of *F. mandschurica*, *F. nubicola, F. pentaphylla*, and *F. chiloensis* showed higher peak levels of burst. In both genomes of *F. daltoniana* and *F. nilgerrensis*, a *gypsy* transposition burst occurred approximately 4 to 5 million years ago (Figure 5B). Compared to other strawberry genomes, the *gypsy* transposition burst in *F. mandschurica*, *F. nubicola*, *F. vesca*, *F. pentaphylla*, and *F. chiloensis* occurred more recently (about 300,000 years ago) and achieved higher peak levels.

#### 2.2.5. Recent Transposition Burst of LTR Retrotransposons

Since all strawberry genomes share a recent transposition burst of LTR-RTs within the last million years, we extracted LTR-RTs inserted during this period for further analysis. The LTR-RT copies were classified into different families using TEsorter [39], and the percentage of each LTR-RT family in the total LTR-RT copies in each species was calculated and used for clustering (Figure 6 and Appendix A). The clustering grouped *F. pentaphylla*, *F. nubicola*, *F. chiloensis*, and *F. x ananassa*, the last four species diverged from the *Fragaria* genus, into one cluster, whereas the others were grouped into another cluster, which is in concordance with the phylogenetic tree [20,21,25], suggesting that closely related strawberries tend to share similar LTR-RT profiles.

The higher the percentage, the more likely an LTR-RT family is one of the primary groups that was involved in recent transposition burst. Both the *Ale* family, a member of the *copia* retrotransposon, and the *Athila* family, a member of the *gypsy* retrotransposon, were found to have high copy numbers in all strawberry species, suggesting that they have had high transposition activity in the past million years. The *Bianca* family in the *F. mandschurica*, *F. nubicola*, and *F. nubicola* genomes; the *Tekay* family in the *F. mandschurica* and *F. vesca* genomes; and the *Athila* family in the *F. chiloensis* genome exhibited a significantly higher abundance (Figure 6). The massive number of copies of these superfamilies suggests that they were one of the primary drivers of the recent LTR-RT transposition burst in those genomes.

### 2.3. Evolution History of TEs in Cultivated Strawberry

#### 2.3.1. Comparative Analysis of the TE Contribution between Cultivated Strawberry Subgenomes and Their Diploid Ancestor Genomes

The cultivated strawberry (*F. x ananassa*) is an allo-octoploid that originated from four diploid ancestors [19,20,33]. However, although three of these ancestors have high-quality genomes that were assembled to the chromosome level [21,22,25], only a fragmented genome assembly is available for *F. nipponica* [35]. Therefore, we only identified TEs in the genomes of three extant diploid ancestors: *F. viridis*, *F. vesca*, and *F. iinumae*. When comparing the TE content in each subgenome of *F. x ananassa* with that in the corresponding ancestor genome, we observed significant differences in the distribution of TE superfamilies. As evident in Figure 7B and Appendix A, the proportion of TEs in subgenome *Camarosa vesca* (the subgenome that likely originated from *F. vesca*—please see Materials and Methods for the naming convention of the four subgenomes) was lower (50.48%) compared to that in the other three subgenomes (56.95%, 57.54%, and 57.59%, respectively). The LTR-RT content in subgenome *C. vesca* was notably lower than that of other subgenomes, with *gypsy* constituting only half of that found in other subgenomes. The class II TE copy number of subgenome *C. vesca* was slightly higher than that in other subgenomes.

By comparing the subgenomes and their corresponding extant diploid ancestors, differences in the landscape of TEs between cultivated strawberry and its diploid ancestors became evident (Figure 7). There were notable distinctions between the TE superfamilies of subgenome *Camarosa viridis* and the *F. viridis* genome. Aside from LTR-RTs, most elements in the *C. viridis* subgenome exhibited lower levels compared to the *F. viridis* genome. Additionally, the *C. viridis* subgenome had a significantly higher total TE content than that of *F. viridis*, with respective percentages of 57.54% and 51.83%. It is also worth noting that the content of the gypsy superfamily of LTR-RTs in subgenome *C. viridis* was nearly twice that of *F. viridis*. Similar findings were observed in subgenome *C. vesca* (50.48%) and the *F. vesca* genome (52.56%), with the exception of LTR-RTs. This difference may be attributed to TE activation during the polyploidization process of cultivated strawberry [40]. Furthermore, the proportion of all elements of subgenomes *C. iinumae*, *C. viridis*, and *C. nipponica* was slightly lower across TE superfamilies compared to the TE content of *F. iinumae*.

#### 2.3.2. Evolution History of LTR-RTs in Cultivated Strawberry

To analyze the LTR-RT transposition history in the subgenomes of cultivated strawberry, we created a distribution curve of LTR-RT insertion times for each of the four subgenomes, as illustrated in Figure 8A,B. Our results indicate that the transposition histories of the LTR-RTs in the subgenomes of cultivated strawberry are similar. We also found a significant transposition burst of both *copia* and *gypsy* elements in the four subgenomes that occurred approximately one million years ago. Among the subgenomes, *C. vesca* exhibited the highest peak level. Additionally, a major transposition burst of *copia* that occurred further back in time was observed in the subgenomes *C. iinumae* and *C. vesca*. The subgenome *C. vesca*, which has a closer genetic distance to extant diploid ancestor *F. vesca*, showed the most significant variation. On the other hand, *F. vesca* displayed a more recent and higher peak level during the recent major *gypsy* transposition burst.

Additionally, the diversity of the LTR-RT superfamilies in the subgenomes of cultivated strawberry and their corresponding diploid ancestors over the past two million years was compared (refer to Figure 6 and Figure 7). *F. vesca*, the diploid ancestor with a distinct *gypsy* insertion time distribution compared to subgenome *C. vesca* [18,19,20,21], exhibited a substantially greater abundance of the *Tekay* family (Figure 6), suggesting that the *Tekay* family experienced an independent transposition burst in diploid strawberry *F. vesca* after the formation of cultivated strawberry.

#### 2.3.3. Shared TEs in Cultivated Strawberry Genome and the Ancestor Genomes

It has been proposed that the cultivated strawberry (*F. x ananassa*) is an octoploid that originated from four diploid ancestors [19,20,33]. Although TE content in the strawberry genomes has been subject to constant changes during evolution, the TE profiles in each subgenome should have high similarity to that in the corresponding ancestor diploid genome [41,42]. To this end, we examined the shared intact LTR-RTs, TIRs, and Helitrons in the genome of cultivated strawberry and the genomes of three proposed diploid ancestors (*F. nipponica* was not in this analysis due to lack of a high-quality genome assembly) using the shared sequences of the pan-genome TE library. As shown in the Venn plot, the number of intact TEs shared exclusively between the genome of *F. viridis* and cultivated strawberry was only 165, which is significantly lower than the 431 in *F. vesca* and the 259 in *F. iinumae* (Figure 9). The low level of shared TEs between the *F. viridis* genome and the cultivated strawberry genome implies that *F. viridis* may not be a true diploid ancestor of cultivated strawberry.

We further counted the copy numbers of intact TEs shared by the subgenomes of cultivated strawberry and three proposed diploid ancestor genomes (Table 2). *F. viridis* and the four subgenomes shared a low copy number of intact TEs (80–120), and the highest copy number of shared TE was found between *F. viridis* and the subgenome *C. vesca*, which is the subgenome that originated from *F. vesca*. In contrast, the numbers of shared TE copies between the *F. vesca* genome and the four subgenomes of cultivated strawberry range from 127 to 516, with the highest copy number coming from the *F. vesca* genome and the *C. vesca* subgenome, which originated from *F. vesca*. Similarly, the numbers of shared TE copies between the *F. iinumae* genome and the four subgenomes of cultivated strawberry range from 73 to 243, with the highest copy number coming from the *F. iinumae* genome and the *C. iinumae* subgenome, which originated from *F. iinumae*. These results suggest that *F. vesca* and *F. iinumae* are true diploid ancestors of cultivated strawberry but that *F. viridis* may not be a true diploid ancestor.

## 3. Discussion

### 3.1. The Contributions of TE Amplification and DNA Removal to Genome Size

Transposable elements (TEs) and other repetitive sequences are crucial components of eukaryotic genomes, which are instrumental in driving changes in genome size [8,43]. The contribution of TEs to genome size is variable in different eukaryotic lineages [44]. Previous reports indicated that TEs exhibit significant species specificity [45], however, general TE databases lack specificity for the *Fragaria* species [46]. Here, we built a pan-genome TE library based on ten high-quality genomes in the *Fragaria* genus, which should serve as an important resource for TE detection and investigation in strawberry genomes. The pan-genome TE library approach can help improve the TE annotation in each genome, especially for TEs that have very low copy numbers in each genome. Comparison of TEs in multiple closely related species can give us a clear picture of TE evolution in these species, and including TE markers in genome-wide association studies may help us find the cause for certain phenotypic changes. As the number of released plant genomes continues to increase at an unprecedented speed, a pan-genome TE analysis can be applied to many other plant lineages.

This study suggests that TEs are a major force in shaping the *Fragaria* genomes. A strong positive correlation between TE content and genome size was observed in the *Fragaria* species, suggesting that the amplification efficiency of TEs is an important factor that influences the genome size. Similar results have also been reported in previous studies for rice [47], Cucurbitaceae [16], Noctuidae [48], arthropods [43], hydras [49], and vertebrates [44]. The half-life rate of TEs serves as a proxy for DNA removal, and a larger half-life rate indicates a lower rate of TE DNA removal. Our analysis of half-life of LTR-RTs suggests that the rate of DNA removal is variable in different strawberry genomes and is also a contributor to genome size. This feature is particularly evident in birds due to the extremely high rate of DNA removal caused by extreme selective pressures [50]. Further investigation of the effects of environmental conditions on strawberry genome evolution may provide additional evidence.

### 3.2. The Role of TEs in Strawberry Genome Evolution

TEs appear to have played a crucial role in the evolution of *Fragaria* genomes. *F. iinumae* is considered the earliest diploid strawberry diverged from other species in the *Fragaria* genus (approximately 7.94 million years ago) [25]. The TE composition of *F. iinumae* differs significantly from that of other strawberry genomes examined in our study. Its genome showed the highest content of LTR-RTs (34.19%) and the lowest content of DNA transposons (22.3%). The transposition bursts of LTR-RTs in the strawberry genomes mainly occurred in the past 2 million years, which is in line with previous findings in other Rosaceae species such as apple [51], peach [52], and plum [15]. The large number of LTR-RTs inserted in the past million years suggests that the evolution of the *F. iinumae* genome was primarily driven by the species-specific transposition explosions of three *gypsy* retrotransposon families (*Athila*, *Ogre*, and *Retand*). This type of genome evolution, caused by specific LTR families, has also been found in the *F. vesca* genome. A differential transposition burst event one million years ago was discovered between the *F. vesca* and the *Camarosa vesca* subgenomes of cultivated strawberry. In the *F. vesca* genome, the *Tekay* family, a *copia* retrotransposon family, has undergone massive transpositions, leading to this different transposition burst. A similar example was found in the genome of hydra [49], where the significant embedding of the *CR1* family of LINE retrotransposons has resulted in genomes three times larger in brown hydras compared to green hydras. Thus, the roles of TEs in shaping plant genomes cannot be ignored.

The copy number of different LTR-RT families can serve as an indicator of the genetic divergence among *Fragaria* species. We utilized TEsorter to sort LTR-RTs into different families and found that the LTR-RT family composition could be used to cluster the species of the Fragaria genus into two distinct groups (Figure 6). One group comprised the species *F. pentaphylla*, *F. nubicola*, *F. chiloensis*, and *F. x ananassa*, which recently diverged from in the *Fragaria* lineage [20,25], whereas the other group comprised species that diverged earlier. Moreover, *Fragaria* species that are close in this clustering usually also have close genetic distances, suggesting that the LTR-RT family composition is a reliable indicator of genetic relationships.

### 3.3. New Perspective on the Diploid Ancestors of Cultivated Strawberry

This study compares the TEs present in cultivated strawberry and their potential diploid ancestors. The likelihood of a diploid species being an ancestor to the cultivated strawberry can be investigated from the TE composition perspective. The cultivated strawberry is an allo-octoploid that was derived from diploid ancestors and was originated through the hybridization of two octoploids, *F. chiloensis* and *F. virginiana* [20]. Early reports have identified *F. vesca* and *F. iinumae* as diploid ancestors [18,19]. However, there exists an open debate on the other two diploid ancestors [20,22,23,33]. The debate centers on whether *F. viridis* and *F. nipponica* are the remaining two diploid progenitors [20,22,23]. Here, we analyzed and compared the TE composition profiles of the subgenomes of cultivated strawberry and the genomes of proposed diploid ancestors. Similar TE content characteristics were discovered in *F. iinumae* and in three subgenomes of cultivated strawberry, including the subgenome *Camarosa iinumae*, the subgenome *C. viridis*, and the subgenome *C. nipponica*. Only the subgenome *Camarosa vesca* possessed the same TE content characteristics as *F. vesca*. However, *F. viridis* showed different TE content characteristics than all other cultivated strawberry subgenomes. Furthermore, *F. vesca* and *F. iinumae* contributed to the highest copy number of shared TEs in the corresponding subgenomes in cultivated strawberry. The contribution of *F. viridis* to shared TEs in cultivated strawberry was significantly lower than those from *F. vesca* and *F. iinumae*, and a similar copy number of shared TEs was distributed among four cultivated strawberry subgenomes. This fact suggests that *F. vesca* and *F. iinumae* are true ancestors of cultivated strawberry, whereas *F. viridis* is not. Because a chromosomal-level genome assembly is not available for *F. nipponica*, it was not included in our analysis, and thus, we cannot provide more evidence on whether it was a possible diploid ancestor. More extant diploid strawberry genomes need to be investigated to determine other true ancestors of the cultivated strawberry. Additionally, machine learning (ML) may be employed for accurate annotation of TEs, as it has already had a wide range of applications in genetics and genomics, such as identifying gene-splicing sites, promoters, and enhancers [53,54]. Although most TEs are non-functional by themselves, their proximity or association to functional genes may have functional consequences. Including TEs as genetic markers in machine learning models may increase the power of predictive breeding or genomic selection, which is an important method for crop improvement under diverse environments [55,56,57].

## 4. Materials and Methods

### 4.1. Genome Datasets

The genomes of ten species in the genus *Fragaria*, assembled at the chromosome level, were selected for the detection, annotation, and analysis of TEs. Details of the genome assembly information are provided in Appendix A. The genomes of nougou strawberry (*F. iinumae*) [22], huangmao strawberry (*F. nilgerrensis*) [28], Himalaya strawberry (*F. nubicola*) [33], woodland strawberry (*F. vesca*) [21], green strawberry (*F. viridis*) [25], cracked strawberry (*F. daltoniana*) [58], northeast strawberry (*F. mandschurica*) [31], five-leaf raspberry (*F. pentaphylla*) [25], Chilean strawberry (*F. chiloensis*) [34], and garden strawberry (cultivated strawberry, *F. x ananassa*) [20] were downloaded from the Genome Database for Rosaceae (GDR) [58]. The sequencing material used for the cultivated strawberry *F. x ananassa* genome was the “*Camarosa*” cultivar. Following the convention used by Liston and colleagues [23], the four subgenomes were referred to as *Camarosa vesca*, *C. iinumae*, *C. nipponica*, and *C. viridis* in accordance with the name of the corresponding proposed diploid progenitor, namely, *F. vesca*, *F. iinumae F. nipponica*, and *F. viridis*, respectively [20].

### 4.2. Construction of the Pan-Genome TE Library for Fragaria Species

A pan-genome TE library was constructed by combining de novo and Repbase [46] annotations for ten *Fragaria* species. TEs of each *Fragaria* species were identified using a de novo approach with EDTA (v2.1.0) [36], which uses tools such as LTR_retriever [59], TIR-Learner [60], and HelitronScanner [61] to accurately identify TEs, in addition to calling on RepeatModeler (v2.0.1) [62] to increase the sensitivity of TE identification (--sensitive 1). Subsequently, the TE sequences of each *Fragaria* species were merged to generate a draft TE database. Simple repeat sequences, satellite sequences, and sequences less than 80 bp in length were discarded. Redundancies in TE sequences were removed using make_panTElib.pl [63], a utility tool from EDTA [36], to obtain the *Fragaria* pan-genome TE library. The de-duplication parameters of make_panTElib.pl were “-miniden 80 -mincov 80”. The unknown sequences in pan-genome TE library were classified into superfamilies using DeepTE [64].

### 4.3. Genome Masking

The pan-genome TE library was used to mask the genomes of ten *Fragaria* species with EDTA [36], the “--anno 1 --step anno” was used to call on the masking software RepeatMasker (http://repeatmasker.org, accessed on 3 September 2020, version 4.1.1), and the identification step was skipped.

### 4.4. Kimura Distance-Based Propagation Activity Analysis

To estimate the propagation activity of transposable elements during the evolutionary history of *Fragaria*, we performed a Kimura-based analysis of the propagation activity of the TE subfamilies. The Kimura distance between each TE and the corresponding consensus sequence in the pan-genome TE library was calculated in alignment files using calcDivergenceFromAlign.pl, a utility tool from RepeatMasker.

### 4.5. Estimate the Insertion Time of LTR Retrotransposon

The insertion time of an intact LTR retrotransposon can be calculated using the genetic distance between two LTR fragments and the neutral mutation rate. The procedure was implemented as scripts in the EDTA package [36]. The Kimura distance of LTR retrotransposons was transformed into an insertion time with the equation T = K/2r, where r is the strawberry neutral mutation rate estimate, which was previously estimated to be 2.8 × 10^−9^ [25], and K is the Kimura 2 parameter divergence of two LTR sequences of each intact LTR retrotransposon.

### 4.6. Estimate the Half-Life Rate of LTR Retrotransposon

To estimate the half-life rate at which LTR retrotransposons are removed from the strawberry genome, we divided the estimated insertion times of full-length LTR retrotransposons from each genome of *Fragaria* into bins of 500,000 years. Assuming that LTR retrotransposons are removed from the genome at a constant rate, the distribution of insertion times can be represented by an exponential function with a constant half-life rate. To estimate the average half-life of LTR retrotransposons, an R (v4.1.1) script was used to fit an exponential function representing the relationship between insertion time and number of LTR retrotransposons to the observed data by minimizing the sum of the distance between the calculated and the observed values for each bin.

## 5. Conclusions

Transposable elements are an important part of the genome and play a crucial role in plant genome structure and function [5,65]. In this study, we conducted a comparative analysis of TEs in ten *Fragaria* genomes to enhance our understanding of the TE composition of each genome and the roles they played in strawberry genome evolution and function. The linear relationship between the half-life of intact LTR-RTs and genome size suggests that, in addition to the amplification efficiency of TEs, the rate of DNA removal is also an important force in determining strawberry genome size. The copy number of LTR-RT families over the last million years reflects the genetic distance among *Fragaria* species. Comparisons of TEs in the genomes of cultivated and diploid strawberries suggest that *F. vesca* and *F. iinumae* are diploid ancestors of cultivated strawberry but that *F. viridis* may not be. More high-quality strawberry genomes are necessary to determine additional diploid ancestors of cultivated strawberry.

## Figures and Tables

**Figure 1 ijms-24-16935-f001:**
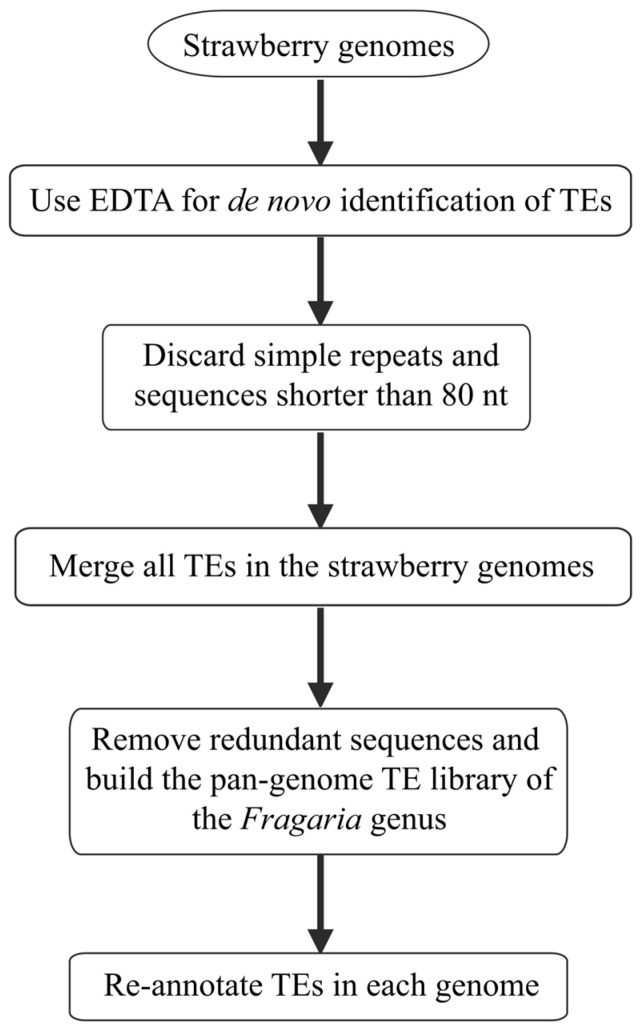
The pipeline for building a pan-genome TE library and annotating TEs in the *Fragaria* genomes.

**Figure 2 ijms-24-16935-f002:**
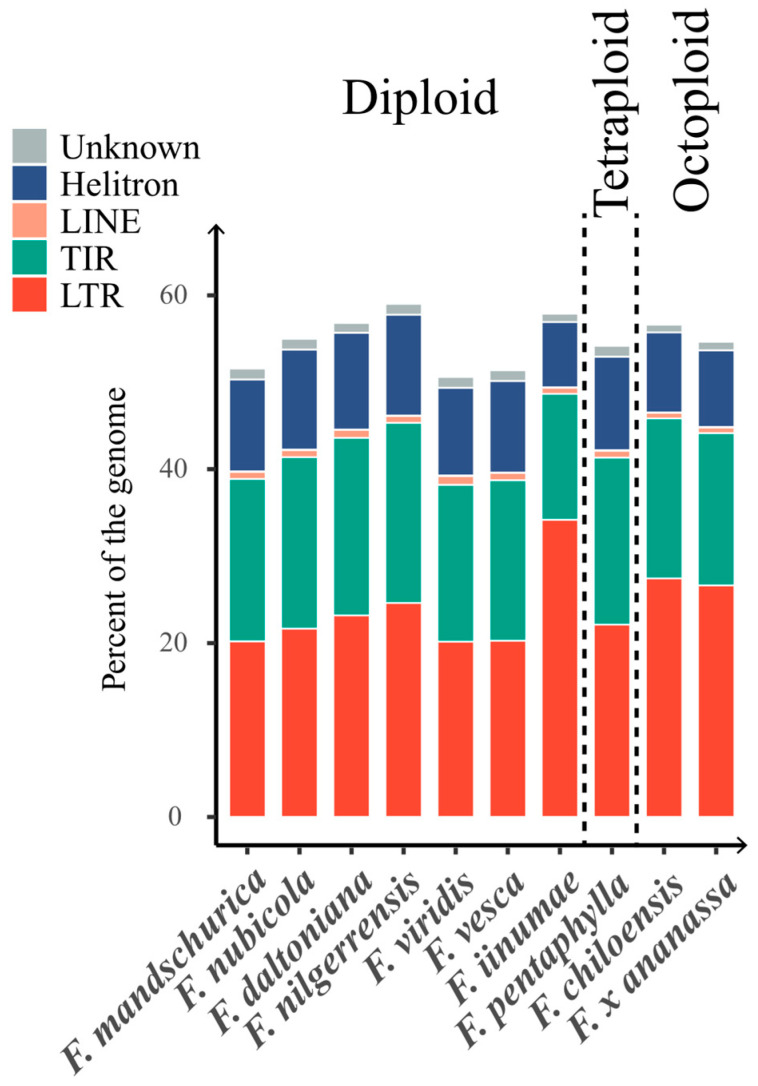
Percentages of genome occupied by TEs in the different *Fragaria* genomes studied. The genome percentages of LTR (long terminal repeat) and LINE (long interspersed nuclear element) retrotransposons; TIR (terminal inverted repeat) and Helitron DNA transposons; and unclassified elements (Unknown) were estimated with EDTA. For clarity, DIRS (*Dictyostelium* intermediate repeat sequence) sequences were included in LTR retrotransposons, and PLE (Penelope-like element) sequences were included in LINE elements.

**Figure 3 ijms-24-16935-f003:**
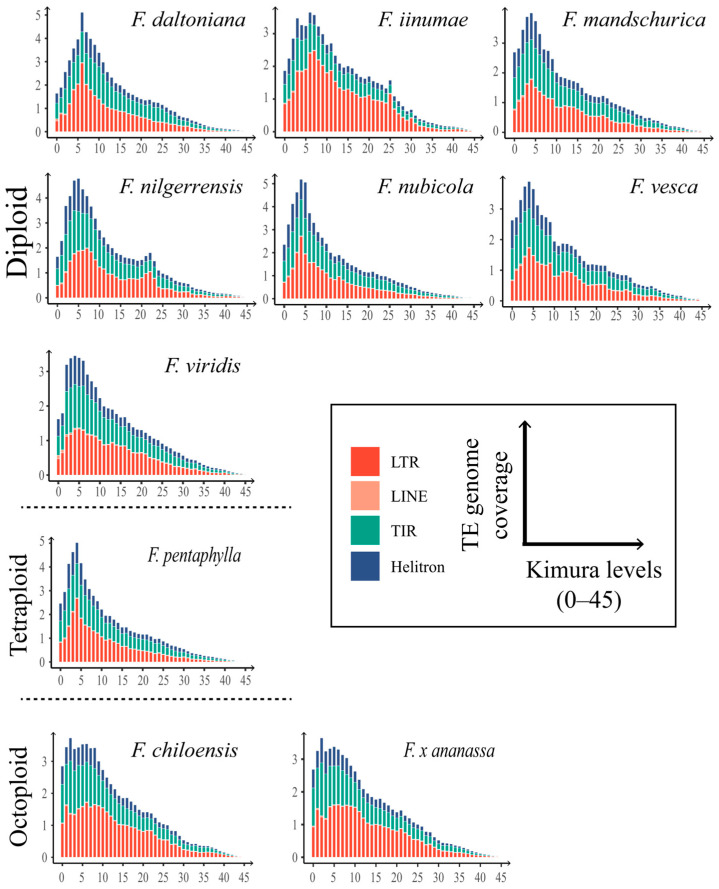
Kimura distance-based divergence analysis of TEs in *Fragaria* genomes. These graphs represent the genome coverage for different types of TEs (LTR and LINE retrotransposons; TIR and Helitron DNA transposons) in the analyzed genomes, clustered according to the Kimura distance of TE copies and their corresponding consensus sequences. For clarity, DIRS sequences are included in LTR retrotransposons and PLE elements in LINE elements.

**Figure 4 ijms-24-16935-f004:**
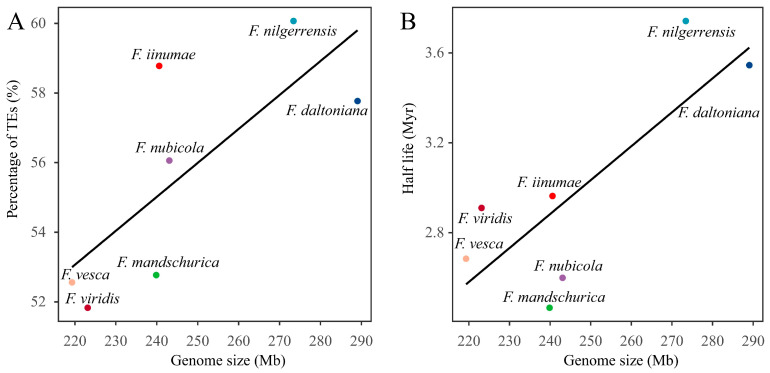
(**A**) Correlation between genome size and content of TEs) and (**B**) between genome size and the half-life rate of LTR-RTs in diploid strawberry genomes.

**Figure 5 ijms-24-16935-f005:**
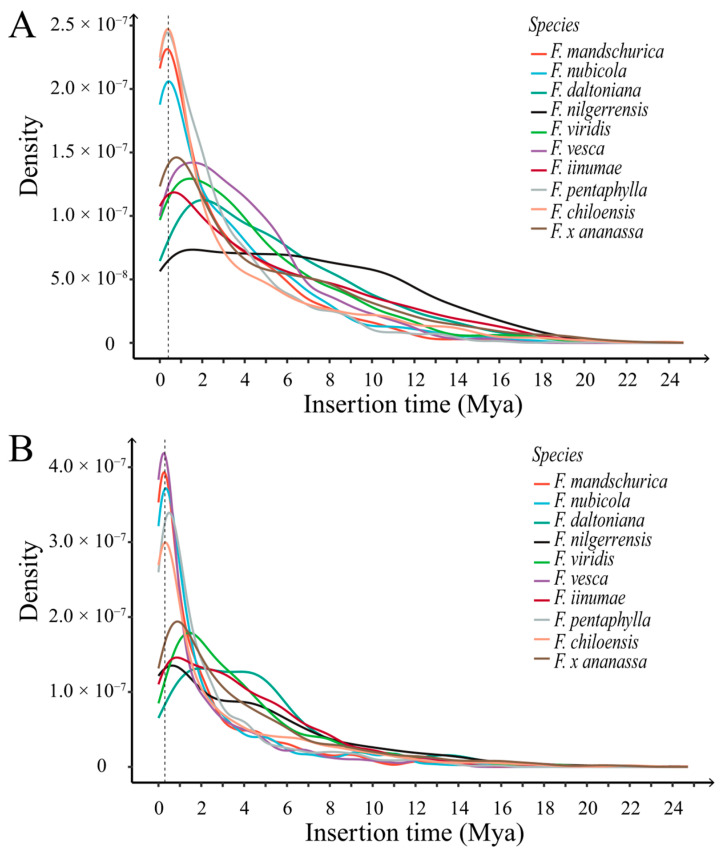
Density plot of LTR-RT insertion times in the *Fragaria* genomes. Colored lines represent different strawberry species. (**A**) Density plot of *copia* retrotransposon insertion time in the *Fragaria* genomes (dashed line at x = 0.4 Mya). (**B**) Density plot of *gypsy* retrotransposon insertion time in the *Fragaria* genomes (dashed line at x = 0.3 Mya).

**Figure 6 ijms-24-16935-f006:**
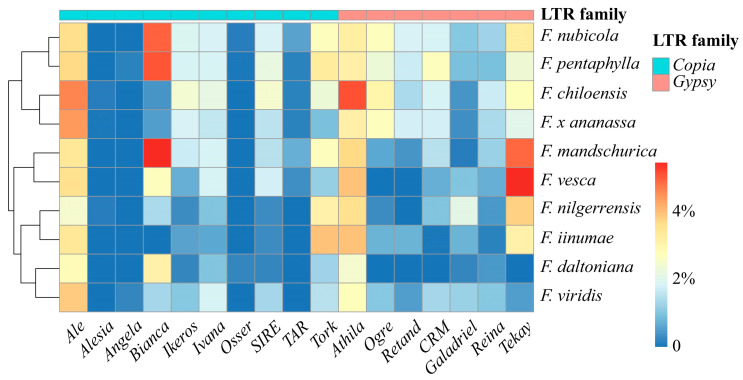
Diversity and abundance of LTR retrotransposon superfamilies that have transposed within the last 1 million years. The family of LTR retrotransposons is identified by TEsorter [39]. Percentages of LTR-RT superfamilies in total LTR-RTs of Fragaria genomes are represented by different colors. Percentage values were log 2 transformed.

**Figure 7 ijms-24-16935-f007:**
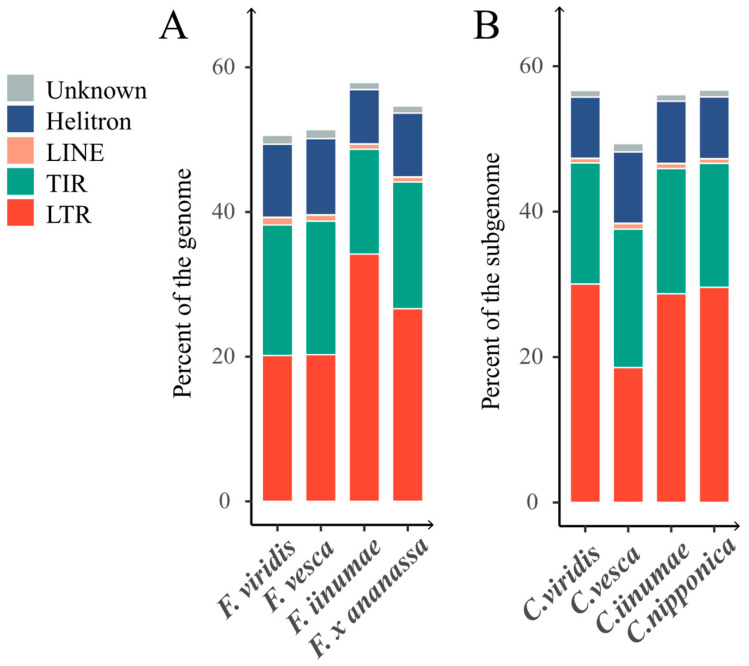
Percentages of TEs in extant diploid *Fragaria* ancestor genomes and the corresponding cultivated strawberry subgenomes. (**A**) Percentages of different types of TEs in the genomes of extant diploid ancestors. (**B**) Percentages of different types of TEs in the subgenomes of cultivated strawberry. For clarity, DIRS sequences were included in LTR retrotransposons and PLE elements were included in LINE elements.

**Figure 8 ijms-24-16935-f008:**
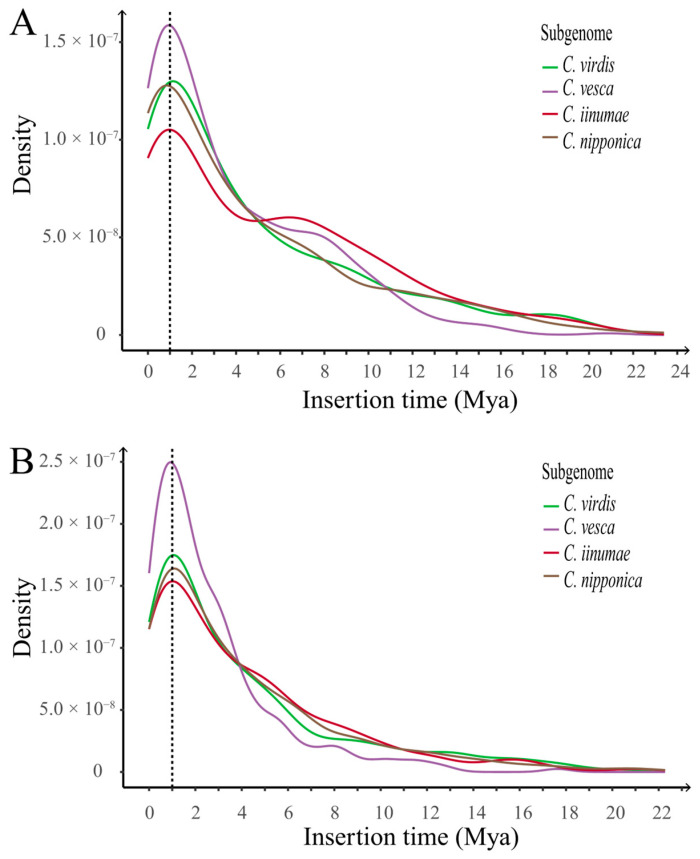
Density plot of LTR-RT insertion times in the subgenomes of cultivated strawberry. Colored lines represent different subgenomes. (**A**) Density plot of *copia* retrotransposon insertion times in the subgenomes (dashed line at x = 1 Mya). (**B**) Density plot of *gypsy* retrotransposon insertion times in the subgenomes (dashed line at x = 1 Mya).

**Figure 9 ijms-24-16935-f009:**
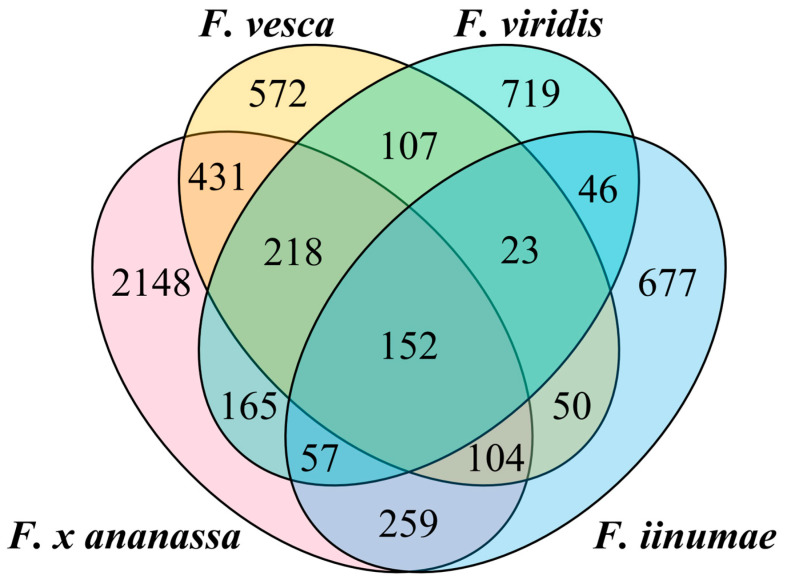
Venn diagram for intact TEs shared between cultivated strawberry and three proposed diploid ancestors.

**Table 1 ijms-24-16935-t001:** Summary of the pan-genome TE library of *Fragaria* species.

Type of TE		Number of Sequences
Class I		8931
LTR		8539
	*Copia*	3002
	*Gypsy*	5255
	Unknown	282
NonLTR		392
	LINE	320
	SINE	46
	DIRS	3
	PLE	23
Class II		16,154
	TIR	13,945
	CACTA	3928
	Mutator	5007
	P	2
	PIF-Harbinger	1408
	Tc1-Mariner	517
	hAT	3083
	Helitron	2209
Unknown		1141
Total		26,226

**Table 2 ijms-24-16935-t002:** Number of TEs shared by the subgenomes of cultivated strawberry and three proposed diploid ancestors.

Subgenome	Total	*F. viridis*	*F. vesca*	*F. iinumae*
*Camarosa viridis*	1639	85	127	157
*Camarosa vesca*	2136	120	516	73
*Camarosa iinumae*	1855	82	139	243
*Camarosa nipponica*	1824	80	151	154

## Data Availability

Data are contained within the article.

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
