# Peer review of "Comparative Analysis of Transposable Elements in Strawberry Genomes of Different Ploidy Levels"

_ijms, 2023, doi:10.3390/ijms242316935_

Round 1

Reviewer 1 Report

Comments and Suggestions for Authors

Ms ID: ijms-2713985

After careful review of the ms entitled “Comparative analysis of transposable elements in strawberry genomes of different ploidy” (ID: ijms-2713985), I am very pleased to recommend it for publication after the suggested revisions.

This reviewer found this ms very interesting in the field of comparative genomics of pant species. This work successfully characterized TEs as important evolutionary factors in the Fragaria genus.

Nevertheless, in order to improve the ms, this reviewer recommends the authors to perform a deep revision of the English language, removing inaccuracies and repetitions. These deficiencies decrease the value of ms, despite it is relevant.

The main suggested revisions are reported below:

TITLE

Change to “Comparative analysis of transposable elements in strawberry (Fragaria spp.) genomes of different ploidy

INTRO

L30: check English language for “a mobile”, remove “a”

L32: add ref. [3] after “plant genomes”

L38: the first time in the main text, define the acronym LTR-RT (see L18)

L37-39: check English language

L48-50: explain better the concept

L51: remove “efforts”

L52: remove “elements”, there is TE(s)

L55: strawberry is not a berry, but a “false fruit”, also known as pseudofruit or pseudocarp.

L60,63: avoid repetitions of “However

L75: use verb and words in plural form for referring to TEs

L78: change “may function” to “are involved”

L80: check “function”, and explain better the concept

L89: check “serve”, use “served” for ancestors

L89: use comma after F. viridis

L92: check “service”

RESULTS

L101: specify if the values refer to aploid genomes

L109: use italics for Fragaria, and specify that Number refers to the number of sequences

L111: check English language, remove “Using” and “, it”

L120: specify “higher end”

L121: remove “significantly” (a statistical analysis was not performed) and change “reduce” to “modified” (neither reduction, neither increase)

L134: specify “comparable”: compared to what if the other values are higher

Figure 1A does not exist

Figure 2: increase resolution, the number of X-axis are not readable

L170: check English language, remove “and”

L186: use correctly italics

L215, 217: check English language, use singular form for “in each Fragaria genomes” and “of each LTR-RT families and transposition bursts”

L218: remove brackets for copia and gypsy

L248: check English language and use passive form (was involved)

L255-256: check English language, use “suggests” and add “of” after “drivers”

L266: specify “high quality sequenced genomes”

L272: please, in M&M section, explain why the name Camarosa was used to refer to subgenomes, here for example: Camarosa vesca (the subgenome that originated from F. vesca)

L279: use the acronym TEs

L284: avoid repetitions, remove “respectively”

DISCUSSION

In the Results section, a first discussion is reported and should be associated with the correspondent part of the Discussion section. Then, restructure the Results section, by renaming this section “Results and Discussion” and then merge the Discussion to the corresponding parts of Results.

L356: avoid repetitions in “constitute vital constituents”

L359-360: the concept is not clear, please explain better

L360: add the subject to the verb “built”

L369: change “suggesting “to “suggests”

L395: remove the repetition of “found that”

L416: use the verb “contribute”

L417: avoid using “While” at the beginning of the sentence

L420: change “It…” to “This fact…”

M&M

L436: use the acronym TEs

CONCLUSIONS

L482: use capital letter for “The copy”

L486: change “may be” to “are”

Comments on the Quality of English Language

see above

Reviewer 2 Report

Comments and Suggestions for Authors

The manuscript addresses an important topic regarding the evolutionary history of the cultivated species of strawberry (Fragaria x ananassa) and its ancestors and close wild relatives (CWR) assessing the transposable elements (TEs). The authors constructed a pan-genome transposable elements (TEs) library for the Fragaria genus using published strawberry genomes of different ploidy levels and performed a thorough comparative analysis using a set of software packages to assess the percentage of TEs, the type and amplification over time, as well as their linear relationship with the genome size. The manuscript is well structured and written; however, there are a few issues that the authors are encouraged to elaborate. These are as follows:   

1.      In the Introduction In ln 55-57 and in the Discussion ln 405 is stated that “The cultivated strawberry is an allo-octoploid, derived …”. As this topic is central to the study, the suggestion is to elaborate in both cases (statements) the precise genome composition (i.e., species hybridized, and genomes implicated) of the species F. x ananassa. Thus, in ln 405 the new insight gained from the study should be further emphasized.

2.    In ln 185-186, the family names and the species taxa (i.e. arthropods) should be in regular font.

3.   In the Results section 2.3.1., in ln 272 “Camarosa vesca (the subgenome that originated from F. vesca)” the genus Camarosa is introduced that in association to the species name versa are used to describe a subgenome. The term “Camarosa” usually refers to a cultivated strawberry variety. Thus, it cannot be used as a genus name, unless in the recent literature this is coined, and relevant references should be cited. 

4.    Also in the following text, ln 272-308, the species names C. viridis is mentioned, however, this is a reptile species. Subsequently the names, C. iinumae, C. vesca, C. viridis, C. nipponica are used. Are these strawberry species? Please clarify and if so, provide relevant references for the species names, and how they are related evolutionarily with the Fragaria species.  

5.      It would be nice to show a timeline diagram of the Fragaria species and crosses over time as a schematic representation of the study’s output, in the Discussion section.

Comments on the Quality of English Language

Minor english typo errors should be cerfully checked .

Reviewer 3 Report

Comments and Suggestions for Authors

This work by Lyu et al. pushes forward our understating of transposable elements (TE) in strawberry across ploidy levels. The major advancement of this manuscript is that it carries out a comparative analysis of TE across ten published strawberry genomes of different ploidy levels using a pan-genomic approach. In terms of novelty, the paper addresses a crucial aspect of molecular biology, as the role of TE is essential for genome stability and trait architecture. Understanding TE is of paramount importance. Overall, the manuscript appears to be a valuable contribution to the field of TE, and novel pan-genome has the potential to advance research in this area. Besides, the work is well written, statistically up to date, and highlights key findings. However, before commending acceptance, I have the following major suggestions.

First, the abstract provides a clear and concise overview of the paper's, methods, and findings. It successfully conveys the significance of studying TE. However, it should be more clearly at explicitly describing the main analytical methods in L19.

Second, the last paragraph of the introduction should explicitly enlist (in L88) the research gap before describing in a more concrete way the specific research goals (in L89). Please also close the paragraph with the research hypothesis and the expected results (in L91). This will allow readers focusing on explicit expectations when approaching the reanalysis of strawberry pan-genomes and TE.

In term of methodology, building and utilizing a pan-genome to trace back TE is a major accomplishment. The simplicity and accuracy of the method is desirable, making it a valuable tool for researchers in the field. I would simply recommend presenting diagram as figure 1 that makes reference to the methodology section, since it would clarify the analytical pipeline.

The results are clear and encouraging. I would suggest that figure 1-8 are presented in higher resolution since so far they look pixelated, which limits their inspection and readability.

In terms of the discussion, even though it effectively summarizes the key points, the paper could benefit from a more detailed discussion of the broader implications of the findings and how they contribute to our understanding of TE and genome stability across other plant species, not just the Fragaria genus. Additionally, providing some insight into the caveats of the method (in L421) would be valuable, follow up by a discussion on the potential future directions for research in this area as a novel perspectives section in L422. In this new perspectives section, please also think outside the box by suggesting how explicit machine learning (ML) strategies could help predicting TE and their role in functional genomics (cite Nat Rev Genet 2015 16(6):321-32), but also for other genomic applications (like predictive breeding, refer to and cite Trends Plant Sci 2014 19(12):798-808, Front Plant Sci 2020 11:583323, J Plant Physiol 2021 257:153354, and Genes 2021 12:783). Mind the potential of predictive molecular breeding for TE as the genomic bases of complex polygenic adaptive trait architectures.

Round 2

Reviewer 2 Report

Comments and Suggestions for Authors

The authors have addressed adequately reviewer's suggestions.

Reviewer 3 Report

Comments and Suggestions for Authors

Authors have done a fine job improving this version, which is why I am able to recommend acceptance at the current stage.